# Human exposure to $PM_{10}$ microplastics in indoor air

**Nadiia Yakovenko**[1☯*], **Lucía Pérez-Serrano**[1☯], **Théo Segur**[1], **Oskar Hagelskjaer**[1,2], **Henar Margenat**[2], **Gaël Le Roux**[2], **Jeroen E. Sonke**[1*]

**1** Géosciences Environnement Toulouse, CNRS/IRD/Université de Toulouse, Toulouse, France, **2** Centre de Recherche sur la Biodiversité et l'Environnement, CNRS/INP/Université de Toulouse, Auzeville-Tolosane, France

☯ These authors contributed equally to this work.
* nadiia.yakovenko@get.omp.eu (NY); jeroen.sonke@univ-tlse3.fr (JES)

## Abstract

The ubiquitous presence of airborne microplastics (MPs) in different indoor environments prompts serious concerns about the degree to which we inhale these particles and their potential impact on human health. Previous studies have mostly targeted MP in the 20–200 µm size range, which are less likely to efficiently penetrate into the lungs. In this study, we specifically investigate airborne, indoor suspended MPs in the inhalable 1–10 µm ($MP_{1-10\ µm}$) range in residential and car cabin environments, by using Raman spectroscopy. The median concentration of total suspended indoor MPs for the residential environment was 528 MPs/$m^3$ and 2,238 MPs/$m^3$ in the car cabin environment. The predominant polymer type in the residential environment was polyethylene (PE), and polyamide (PA) in the car cabin environment. Fragments were the dominant shape for 97% of the analyzed MPs, and 94% of MPs were smaller than 10 µm ($MP_{1-10\ µm}$), following a power size distribution law (the number of MP fragments increases exponentially as particle size decreases). We combine the new $MP_{1-10\ µm}$ observations with published indoor MP data to derive a consensus indoor MP concentration distribution, which we use to estimate human adult indoor MP inhalation of 3,200 MPs/day for the 10–300 µm ($MP_{10-300\ µm}$) range, and 68,000 MPs/day for $MP_{1-10\ µm}$. The $MP_{1-10\ µm}$ exposure estimates are 100-fold higher than previous estimates that were extrapolated from larger MP sizes, and suggest that the health impacts of MP inhalation may be more substantial than we realize.

## Introduction

Microplastic (MP) is a ubiquitous pollutant resulting from the global extensive human use of plastic materials since 1950 and the mismanagement of plastic waste [1]. The term "microplastic" refers to plastic particles between 1 µm and 5 mm in size that come in a variety of shapes, and polymer compositions, and can be classified by

**Data availability statement:** All relevant data are within the manuscript and its Supporting information files

**Funding:** ANR-20-CE34-0014 ATMO-PLASTIC ANR-23-CE34-0012 BUBBLPLAST

**Competing interests:** The authors have declared that no competing interests exist

origin as primary (intentionally manufactured MPs) or secondary (MPs generated by the unintentional fragmentation of larger plastic items) [2–4]. Over the past decade, MPs have been detected in outdoor atmospheric aerosols [5–8] and deposition [9–14], in various parts of the world, from urban and highly industrialized areas [9,13] to remote mountainous regions [10,12], the marine boundary layer [7], and indoor environments [15–18]. The ubiquitous presence of MPs in the atmosphere raises many concerns about whether, and to what extent, we are inhaling MPs from outdoor and indoor air, with the latter likely playing the most significant role in human exposure to MPs through inhalation. Recent studies have shown that the concentration of indoor suspended MPs is eight times higher than outdoors [18], and the concentration of indoor deposited MP dust is 30 times higher than outdoors [19]. Given that people in developed nations spend approximately 90% of their time indoors, including 5% in cars [20–23], the potential for inhalation exposure to MPs in indoor environments is significantly higher and warrants attention.

Airborne MPs differ in size, shape, and chemical composition, which determine the mechanism of their interaction with the respiratory system and the nature of potential negative effects. Inhaled particles larger than 10 µm are generally retained in the upper respiratory tract and undergo mucociliary clearance, while particles smaller than 10 µm can penetrate deeper into the lungs. The latter belong to the category of respirable particulate matter (PM) and are usually divided into two categories: $PM_{10}$ (MPs < 10 µm) and $PM_{2.5}$ (MPs < 2.5 µm), which can cause respiratory problems like inflammation and chronic conditions such as bronchitis and asthma [24]. Additionally, MPs can release toxic additives and adsorbed environmental pollutants, that may disrupt endocrine functions and increase risk of various diseases including cancer [25]. This combination of physical and chemical stressors makes MP inhalation a significant public health concern. Therefore, the assessment of human exposure levels is extremely important but remains very challenging due to the novelty of the topic and need for specialized equipment and methods for indoor MP analysis. To date, one of the most commonly employed methods for analyzing indoor MPs is micro-Fourier Transform Infrared (µFTIR) spectroscopy [26]. The µFTIR has a limit of detection (LOD) of 10–20 µm [26,27], and therefore mostly identifies larger MP in indoor air. However, to accurately quantify inhalation exposure and improve the reliability of human health risk assessments, both quantitative and qualitative analysis of MPs within $PM_{10}$ size range are essential, as these particles are small enough to reach the lower respiratory system [28].

Raman spectroscopy (LOD 1µm) is highly effective for targeting smaller particles, making it a valuable tool for detecting MPs in various environments. Several recent studies [17,29–31] have successfully applied Raman spectroscopy to assess indoor suspended $PM_{10}$ MPs, providing valuable insights into this emerging field. However, due to variations in methodology approaches, particle size detection limit, and the types of indoor environments studied, further research is essential to build a more comprehensive understanding of the sources, fate and behavior of $PM_{10}$ MPs in indoor air. To contribute new insights and expand the available dataset for a more accurate and reliable assessment of human inhalation exposure to suspended MPs

in indoor environments, this study focused on: i) developing quantitative and qualitative Raman analysis of suspended indoor MPs down to 1 μm, ii) quantifying MP concentrations in indoor air (MPs/m$^3$) within residential and car cabin environments, and iii) estimating human adult and children indoor MP inhalation exposure (MPs/capita/unit time).

## Materials and methods

### Studied indoor environments

Two types of indoor environments were investigated in this study: a residential environment and a car cabin environment. The residential environment is represented by three apartments (seven samples from bedrooms, home offices, and living rooms) located in different parts of the city of Toulouse (France). No permits were required because the residential and car environments studied are all privately owned by the authors of this study. The car cabin environment is represented by two different cars (old and new, both with polyester fabric seats), from which five samples were collected while driving between the cities of Toulouse and Sigean, Toulouse and Grenoble, as well as within the city of Marseille (all in France). Car cabins indoor environments differ from residential indoor environments, such as homes, offices, and commercial buildings and have a different microclimate. Car cabins are smaller, enclosed and mostly made of synthetic materials such as plastic, which can be potential sources of MPs. Samples were collected between January and May 2023, and details are summarized in S1 Table. Due to the extensive method development needed to use PM$_{10}$ Raman spectroscopy for microplastics, our focus was on analysis quality rather than quantity. As a result, only 16 samples, including blanks, were analyzed in total, which is similar to other studies on indoor MP [17,32,33].

### Sampling of suspended indoor MPs

Indoor suspended MPs were collected through active sampling using 12V and 220V vacuum pumps. The aerosol collection system consisted of a 47 mm PFA Teflon filter holder (Savillex Corp.), with which air was collected through a 10 mm inlet and filtered through PTFE filters (Ø = 47 mm, pore size = 1 μm). The system was coupled with a gas volume meter to measure the sampled air volume. For sampling in the residential environment, windows were kept closed and the filter holders were placed horizontally at a height of 1.6–1.7 m above the floor in living rooms, corresponding to the average human inhalation height [18]. In bedrooms, a height of 0.5 m was chosen to represent the average inhalation height during sleep, based on the bed height in the studied bedrooms. Depending on the volume of collected air ($V_{air}$), the samples were divided into two groups: $V_{air}$ < 3 m$^3$, and samples with $V_{air}$ ≥ 3 m$^3$ (S1 Table). These relatively modest sampling volumes target specifically the more abundant PM$_{10}$ particle size fraction. Studies that use μFTIR to target less abundant large fibers in air require much larger sample volumes, > 1000 m$^3$. Sampling in cars was carried out while driving the car with the windows closed, and a medium flow of outside air into the car via frontal vents. The in-car sampler was battery-powered and the filter holder was attached to the back of the front seat head rest. Further details on residential and car cabin environments are given in S1 Table. All samples were prepared in pre-cleaned individual filter holders under a class 100 laminar flow hood and wrapped in aluminum foil before and after deployment.

### MP extraction prior to analysis

Particles from low volume samples (>3 m$^3$) were directly transferred from the PTFE filter to the Anodisc aluminum oxide membrane filter (Ø = 25 mm, 0.22 μm pore size) without any pretreatment steps. Each PTFE filter was placed in a beaker containing 60 ml of 10% v/v methanol solution and sonicated for 10 min to detach particles from the filter surface. After that, the obtained suspension with the extracted particles was filtered onto an Anodisc filter. To minimize particle loss during transfer, PTFE filter, beaker and the walls of the glass filtration unit were rinsed 3 times with 10% v/v methanol solution into the filtration unit. The Anodisc filter was then placed in a glass petri dish and dried overnight under a class 100 laminar flow hood before analysis.

                                                                                    

Particle extraction from higher volume samples (3–10 m$^3$) included a density separation step aimed at removing inorganic matter that could potentially overlap the MPs on the filter. Density separation was performed using a calcium chloride (CaCl$_2$) solution with a density of 1.4 g/cm$^3$. Sufficient density separation required 7 days and was initiated by transferring particles from the PTFE filter to the CaCl$_2$ solution by sonicating the filter three times (t = 5 min) in CaCl$_2$ solution (50 mL). Every two days, the settled particles were removed from the density separation funnel. To maintain the volume constant, a new CaCl$_2$ solution was then added to the funnel, which was subsequently shaken to mix the contents and allow further particles to settle. After density separation, the samples were filtered onto an Anodisc filter and extensively rinsed with ultrapure Milli-Q water (18.2 MΩ·cm) to avoid CaCl$_2$ precipitation and then rinsed three times with ethanol. Identical to low volume samples, the obtained Anodisc filter was dried and stored in the glass petri dish prior to analysis.

## Raman spectroscopy: morphological and chemical characterization

Raman analysis was conducted at a controlled room temperature (22°C) using a Horiba (Jobin Yvon, France) LabRAM Soleil equipped with a high stability air-cooled He–Cd 532 nm laser diode and Nikon LV-NUd5 100x objective. The laser power was set to 6.3% (5.7 mW). Spectra were collected in the 200–3600 cm$^{-1}$ range using 600 grooves/cm grating with a 100 µm split. The spectra acquisition time was set to 3s with 3x accumulation.

The Raman analysis was performed using automated particle identification (ParticleFinder module in the LabSpec 6 (LS6) software package). A high-resolution visual image of the 1 mm$^2$ filter area analyzed was acquired via the ViewSharp module, using a ±50 µm scan range to focus, and then converted into an 8-bit 0–255 greyscale image in the ParticleFinder module, where contrast parameters are set by the user to visually separate particles from the filter background. After setting all the parameters, the Raman spectra of each particle are collected one by one. In addition to chemical spectra, the ParticleFinder provides information on the particle location and morphological characteristics such as particle size, area, perimeter.

Raw spectra were processed and analyzed using the Spectragryph spectral analysis software V1.2.17d (Dr. Friedrich Menges SoftwareEntwicklung, www.effemm2.de/spectragryph). All spectra were subjected to adaptive baseline correction with a coarseness setting of 15%. Corrected spectra were cross-referenced, using our in-house library, which consists of selected spectra from SLoPP and SLoPP-E [34] and in-house spectra for Polymer Kit 1.0 (Hawaii Pacific University Center for Marine Debris Research: https://www.hpu.edu/cncs/cmdr) as well as plastic collected in the environment. Spectra were considered identified if the spectral hit quality index (HQI) values were higher than the threshold value set for each polymer type (S2 Table).For feasibility reasons, 0.3% (1 mm$^2$; 4,000 spectra; t = 14h) of the total effective area, excluding the outer polypropylene (PP) ring, (283.5 mm$^2$, Ø = 19 mm) of the Anodisc filter was analyzed. The analyzed area corresponds to the center of the filter membrane for all the samples, ensuring that analyses of different samples are consistent and results are comparable and reproducible.

## Quality assurance and quality control (QA/QC)

**Positive control.** For positive controls, red polyethylene (R-PE) beads (Cospheric: https://www.cospheric.com/) of size 10–27 µm were used. Dry R-PE beads were dispersed in a Milli-Q water solution with 1% v/v Tween 20 for better particle dispersion. The positive control samples were spiked with 1–3 mL of R-PE solution (260 ± 36 particles/mL), and tests were performed to evaluate the particle recovery rate through extraction processes, as well as the homogeneity of particle distribution on the Anodisc filter.

**MPs recovery rate.** To assess the recovery rate of the applied protocols, spiked filter samples (n = 3) were optically imaged using the 10x objective of the Raman microscope (50 image mosaic, 2h) and the number of R-PE was counted in the ClickMaster2000 (https://www.thregr.org/wavexx/software/clickmaster2000/) software. Then, spiked samples underwent the same processing steps as real samples. The recovery rate (%) was calculated as the ratio between the number of particles found on the Anodisc filter and the initial number of spiked R-PE:

$$Recovery\,(\%) = \left( \frac{N_{recovered}}{N_{spiked}} \right) x100$$

(1)

Where $N_{recovered}$ is the final number of particles obtained after all processing steps, and $N_{spiked}$ is the initial number of R-PE that were counted on the unprocessed, spiked filters. Recovery rates of 81 ± 3% were observed (n = 5).

**MP distribution on the filter.** The spatial distribution of the spiked R-PE on the Anodisc filter was determined by analyzing their distribution pattern from the center to the edges of the filter. The Anodisc filter effective area (283.5 mm$^2$) was divided into four concentric rings (bins) (with mean distances from the center at 1.25, 3.75, 6.25, and 8.5 mm), and for each of these rings, particle density (MPs/mm$^2$) was determined. The R-PE densities were plotted against the radial distance from the center of the filter (S1 Fig).

**Contamination control.** All sample processing steps were performed under a class 100 laminar flow hood to avoid sample contamination. Operators were equipped with 100% cotton lab coats and nitrile gloves. All sampling tools were made of glass, metal or fluoropolymers (PTFE, PFA) to prevent contact with commodity plastics. Utensils (beakers, petri dishes, filtration units, and density separation funnels) were rinsed with abundant tap water, Milli-Q and ethanol. All glass tools were oven-cleaned before use, for 2 hours at 530°C. All reagents used (Milli-Q, ethanol, methanol and CaCl2 solutions) were filtered through 1.0 µm PTFE and stored in Pyrex bottles (1L) with PTFE screw caps. The squeeze bottles used for Milli-Q and ethanol were made from perfluoroalkoxy alkanes (PFA). All filters were washed with Milli-Q and ethanol on both sides before use for sampling or transfer and manipulated with metal tweezers, which have been cleaned as formerly described. In addition, negative controls (blanks) were generated during sampling and further processing steps to examine potential sample contamination. For the blanks, the PTFE filter was placed in the filter holder without pumping. Filter holders for sampling and their corresponding blank were prepared the same day. The blank filters underwent the same procedure as their corresponding samples.

### Data analysis

To obtain the final concentration of MPs in the air, the obtained Raman data were subjected to the following steps:

1. *Blank correction*, where the number of particles in the samples was adjusted for possible contamination during sample preparation and processing. Polymers found in blanks were subtracted from their corresponding samples based on chemical composition and diameter range (e.g., if the blank presented two particles reported as PE in a [1,2) µm diameter range, these particles were subtracted from the count of PE [1,2) µm class in the sample).

2. *Recovery correction* was performed to account for particle loss during the processing steps and aimed at avoiding overestimation:

$$N_{MP\_recovery\_corrected} = \frac{N_{MPs\_detected}}{recovery\ rate}$$

(2)

Where $N_{MPs-detected}$ is the number of particles identified by Raman analysis and the recovery rate is the percentage of spiked R-PE recovered after all processing steps.

3. *MP number extrapolation to the filter area*. The number of MP particles from the analyzed area (1 mm$^2$) was extrapolated to the entire effective area of the filter (283.5 mm$^2$), following the results from the radial distribution test of R-PE particles across the filter surface (S1 Fig). Because the radial R-PE distribution was constant, indicating homogeneous deposition over the filter, the MP number extrapolation was quasi-linear and proportional to the surface areas.

4. *MP indoor concentration* (MPs/m$^3$) was calculated by dividing the total number of MPs on the sample filter by the air volume pumped through the filter. The standard definition for fine particulate matter, PM$_{10}$, in mass units (e.g., µg/m$^3$)

covers the entire 0–10 µm range. Observation of MP in the 1–10 µm range in abundance/$m^3$ units by microscopy, therefore does not strictly correspond to the full $PM_{10}$ range and definition. We, therefore, report MP observations with the $MP_{10-300 \mu m}$ and $MP_{1-10 \mu m}$ notation, and nanoplastic (NP) estimates with the $NP_{0.1-1 \mu m}$ and $NP_{0.01-0.1 \mu m}$ notation.

5. *MP inhalation* was calculated by multiplying human inhalation rate ($m^3$/capita/day) by MP concentration in air (MPs/$m^3$). The calculations were based on recommended EU default inhalation rates for adults and children [35].

## Results and discussion

### Raman analysis

The Raman analysis involves a two-step process. First, a high-quality optical image of the analyzed area is captured and sent to ParticleFinder™ (Horiba). Here the optical image is processed based on contrast to identify around 3,000 candidate particles on the filter, and subsequently the Raman spectrum of each particle is collected. Micron-sized aerosols, including MP, typically follow a power law size distribution where $MP_{1-10 \mu m}$ sized target particles are most abundant. The amount of time necessary for Raman analysis therefore increases as the target particle size decreases. For analysis of particles down to 1 µm we use a 100x high-magnification objective to obtain high-quality optical images. Consequently, capturing a 1$mm^2$ filter area with the 100x objective and ViewSharp™ (z focus) required capturing 315 optical images to build a mosaic, which takes about 2 hours. In addition to this, the time to collect spectra required 14 hours and on average 3,618 spectra were collected per 1$mm^2$ sample area. The estimated time required for a complete optical image of the analyzed filter (299 x 403 photos) is approximately 669 hours, without the time required to collect the spectra. This makes it impossible to analyze the full area of the filter with the conditions used. Thus, only 0.3% (1$mm^2$) of the total effective area (283.5$mm^2$) of the Anodisc filter was analyzed. The number of particles from this 0.3% filter analyzed was extrapolated to the total effective area of the filter, based on the results obtained from the radial distribution test.

### Radial particle distribution

The radial particle distribution of spiked samples (n = 5) with R-PE was determined by measuring the particle density (MPs/$mm^2$) across four concentric rings on the Anodisc filter (S3 Table, S1 Fig). R-PE particles were observed to be homogeneously distributed over the filter surface (one-way ANOVA; p-value = 0.55). The first concentric ring (d = 5 mm; S = 19.6$mm^2$) accounts for 7% of the filter surface area and 8 ± 3% of the total number of spiked R-PE. We assumed that sample MP processing yielded similar quasi-homogeneous radial MP distribution, and applied a particle number extrapolation to estimate the total number of MP on the filter. Radial distribution analysis is crucial when chemical analysis of particles is not possible on the entire filter surface and only a small area of the filter is analyzed. For example, tests (not shown here) using 1.0 µm PVDF teflon filters, instead of Anodisc filter in the final step resulted in highly non-linear radial MP distributions.

### Blank and recovery correction

In total, 12 samples and 4 corresponding blanks were analyzed. First, the number of particles from the 1$mm^2$ area analyzed for each sample was corrected for blank MP counts by subtracting the corresponding class of polymer and size range from the samples. Contamination accounted for 18% of the total number of particles identified. A total of five types of polymers were detected in the 4 blanks, including PE, PP, polyethylene terephthalate (PET), polyhydroxybutyrate (PHB), and polydimethylsiloxane (PDMS). Although some polymers, such as PE, PET, and PP, were observed in multiple blanks, the detected MPs varied in both polymer types and size range. Overall, no consistent pattern of contamination was observed among the blanks. After blank correction, the number of particles was corrected

for the recovery rate. MP recovery rate through the processing steps has been estimated at 81% (± 4) (n = 3) for a size range of 10–27 µm.

## Indoor MP concentrations

Following blank correction and extrapolation of the results, the median concentration of total suspended MP from all indoor environments (concentrations in 12 samples) is 1,877 MPs/m³ with an interquartile range (IQR), of 478–2,384 (Fig 1, S4 Table). Median residential suspended MP of 528 MPs/m³ (IQR 288–2,487; n = 7) were lower than car cabin MP of 2,238 MPs/m³ (IQR 1,515–2,245; n = 5), likely because of active ventilation inside cars. All residential sampling conditions represented low human activity conditions, except for sample MP15 which had high activity of two persons and resulted in the largest total MP concentration of 34,404 MPs/m³.

## Indoor MP composition

The polymer composition of MPs varies depending on the materials and objects that are an integral part of a certain indoor environment. In total, 10 different polymer types were identified in the environments studied (Fig 2A) and Raman spectral matching is illustrated in Figs 2B, 2C, and S2–S8. Suspended MPs in the investigated apartments were mainly composed of PE (76%) followed by PDMS (6.3%), PA (5.6%), PP (4.2%), PET (4.2%), PHB (1.4%), polybutylene terephthalate (PBT: 1.4%), polystyrene (PS: 0.7%) and acrylonitrile butadiene styrene (ABS: 0.7%). The widespread presence and high concentration of PE can be attributed to it being one of the most commonly produced and utilized polymers globally [36]. The obtained results are consistent with a similar study of suspended indoor fine particulate MPs, where PE accounted for 74% [29], including in the $MP_{1–10\ µm}$ size fraction.

The composition of polymers in cars differs from the composition of polymers in houses. In cars, the most common polymer was PA (25%), followed by ABS (19%), PE (19%), PET (14%), PP (8%), PDMS (8%), PS (3%) and polyvinyl chloride (PVC: 3%). PA and PE are common car upholstery textiles, and ABS and PET are common dashboards and interior panel polymers [37]. We, therefore, suggest that wear and tear of vehicle interior parts made of plastic is a major source of MP exposure to drivers and passengers.

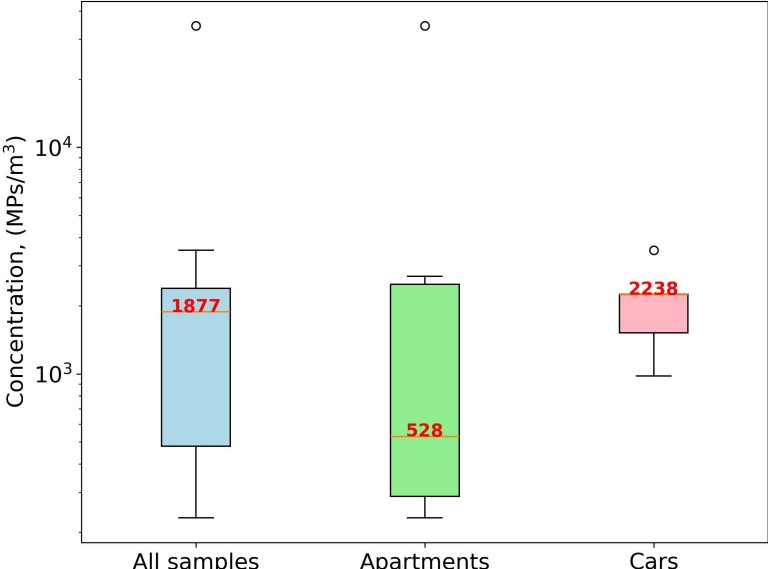

**Fig 1. Indoor total suspended MP concentration (MPs/m³) for all samples (n = 12), apartments (n = 7), and cars (n = 5).**

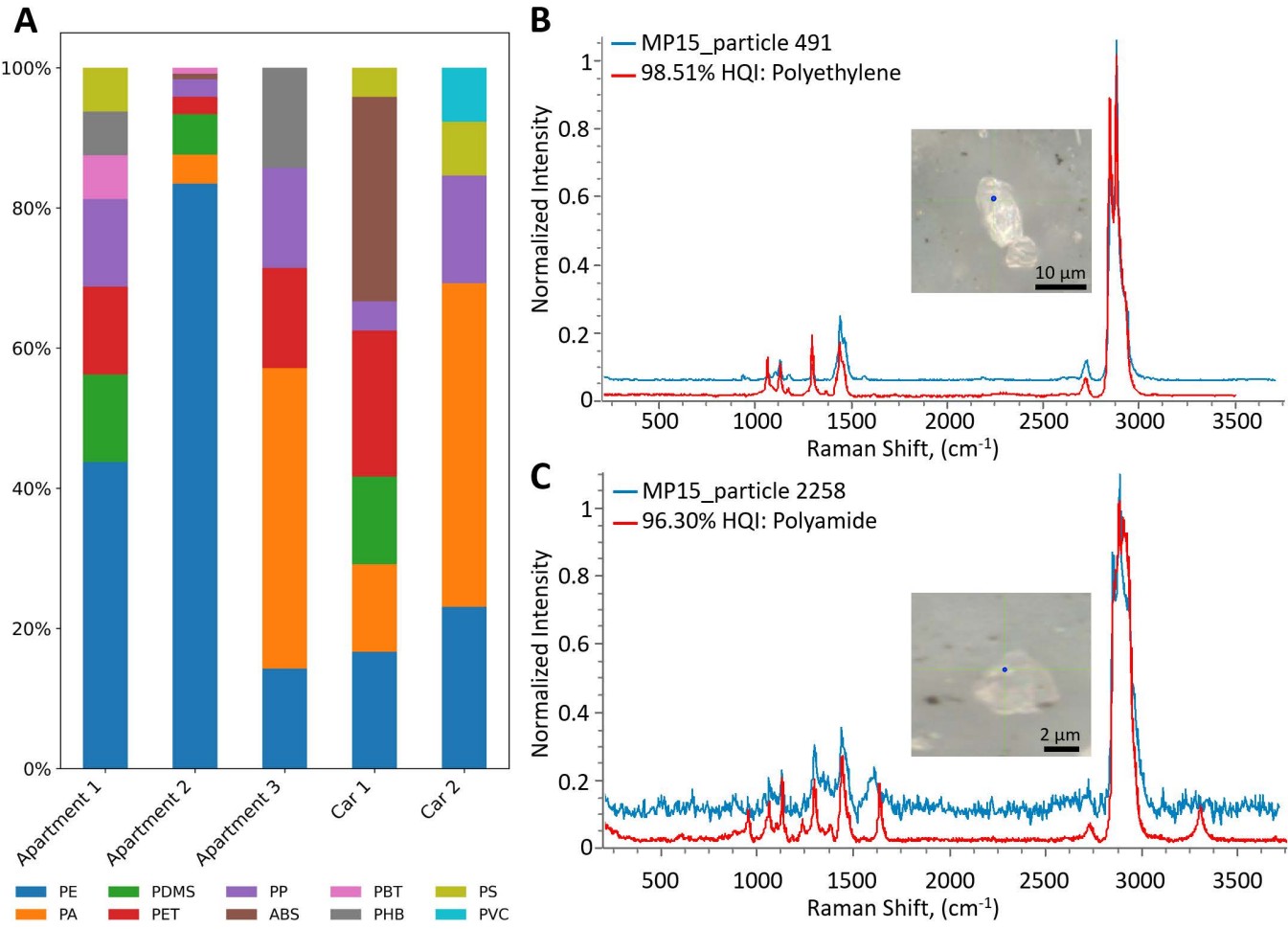

**Fig 2. MP polymer composition in indoor environments.** (A) Total suspended MP polymer composition observed in different indoor environments studied. (B) Raman spectrum of polyethylene (PE) particle (blue) and reference spectrum of PE (red). (C) Raman spectrum of polyamide (PA) particle (blue) and reference spectrum of PA (red).

## Indoor MP morphological characteristics

**Shape.** The analyzed MPs were defined as either fragments or fibers depending on their length-to-width (L/W) ratio. Particles with an L/W ratio > 3 were considered fibers, while particles with an L/W ratio ≤ 3 were classified as fragments [32]. Fragments accounted for 97% of the MPs represented by all 10 types of polymers identified in this study, and the remaining MP were fibers represented by PET, PA, and PP types. Recent studies show a tendency for fragments to dominate small suspended MPs in air [17,18,29,32].

**Size.** The size of MPs in the residential environment was in the range of 1–28 μm and for the car cabin environment from 1 to 15 μm (Fig 3). Larger MP, including fibers, were not identified in this study because the sampling volume (<10 m³) was relatively small and targeted small MP only. The inhalable $MP_{1-10\ \mu m}$ fraction makes up 94% of all detected MPs. MP abundances increase as the particle size decreases (Fig 3) and shows a typical power law distribution, $y = bx^{\alpha}$. S9 Fig summarizes published suspended MP size distributions (log size versus log % relative abundance), with a mean power law exponent of −1.65 ± 0.63 (1σ), that is typical of MP size distributions observed in a range of environments, with α = −1.6 ± 0.5 (n = 19) [38].

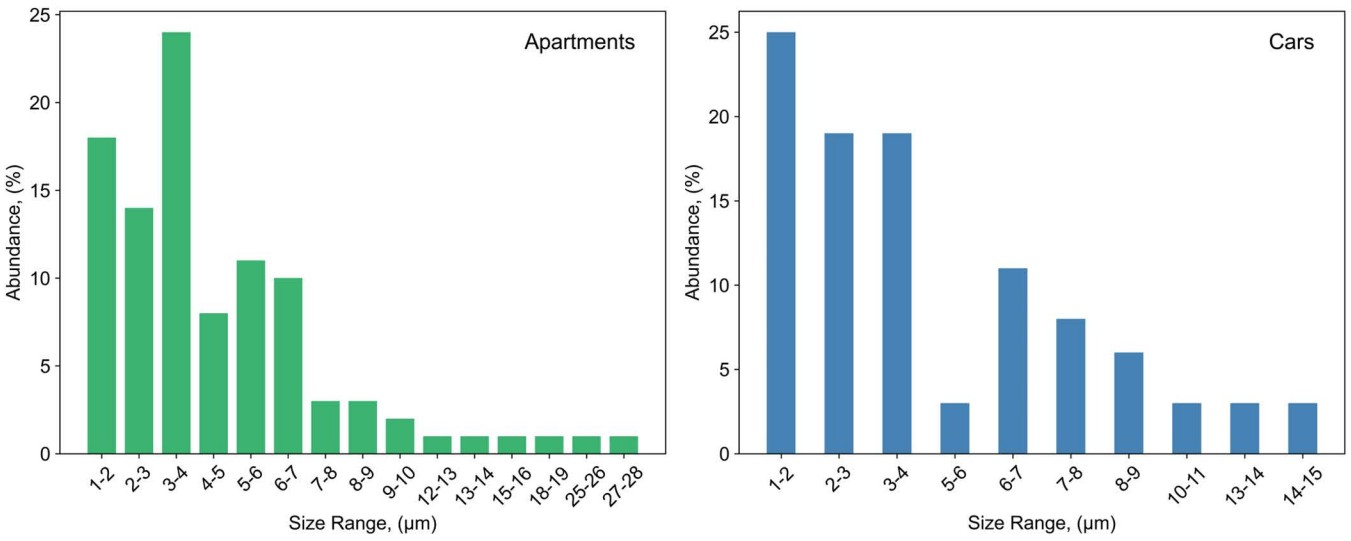

**Fig 3. Observed indoor suspended MP size distribution in apartments (n = 7) and cars (n = 5).**

## Comparison to literature

The presence of MPs in various environments, including households, offices, shopping centers, and public places [17–19,29–33], has been previously identified, demonstrating the ubiquitous presence of MPs in indoor air and indicating exposure by inhalation. Studies are often difficult to compare directly, because different, incomplete ranges of the MP size spectrum (1–5,000 µm) are observed. To compare literature studies, we therefore used the reported relative size distribution (percentage of MP, for a given size range, often 5 µm, 10 µm or larger size bins), and the reported median MP concentrations (MPs/m$^3$), to reconstruct MP concentration distributions for standardized 1 µm wide bins (MPs/m$^3$/µm, see S9 Fig for all data). Using identical size bins, Fig 4 compares the size distribution and indoor suspended MP concentrations from this study to published data. The first thing to notice is that all indoor MP data define a single array in log-log space, suggesting broad coherence between the original observations across the full 1–5,000 µm MP size range. The higher concentration of MPs found in the current work is primarily due to the lower LOD (1 µm), which allows for the identification of more abundant smaller MP compared to µFTIR studies with a higher LOD (10–20 µm) [9,32,39]. Maurizi et al. (2024), using similar Raman microscopy in the MP$_{1–10 \text{ µm}}$ range, reported average indoor MP concentrations ranging from 185 MPs/m$^3$ in new to 548 MPs/m$^3$ in older apartments. These findings are consistent with our results for residential environment MP, which show a median concentration of 528 MPs/m$^3$. Our study is the first to provide data on the presence of suspended MP in the car cabin environment, with a median concentration of 2,238 MPs/m$^3$. Although the median MP concentration in the car cabin environment was higher than in apartments, this difference was not statistically significant (Mann-Whitney U test; p-value = 0.5) due to the high variability in both environments.

Fragmentation of plastic and MP leads to MP particle size distributions that are characterized by exponentially increasing concentrations of the ever smaller fragments that follow a power law. Overall, the standardized data spread for all published suspended MP observations in Fig 4 reflects this inherent power law distribution, but also the true environmental MP concentration differences, and the variability in methodology. Nevertheless, the different studies are highly complementary and define a power law (y = ax$^b$) size distribution with y = 5979x$^{-2.331}$ and r$^2$ of 0.86. We use the linearized form of the power law, log(y) = blog(x) + log(a), with b = −2.331 ± 0.098 and log(a) = 3.776 ± 0.204 (1σ standard errors), and Monte Carlo simulation of 10,000 particle size distributions to derive uncertainty ranges for the indoor suspended MP

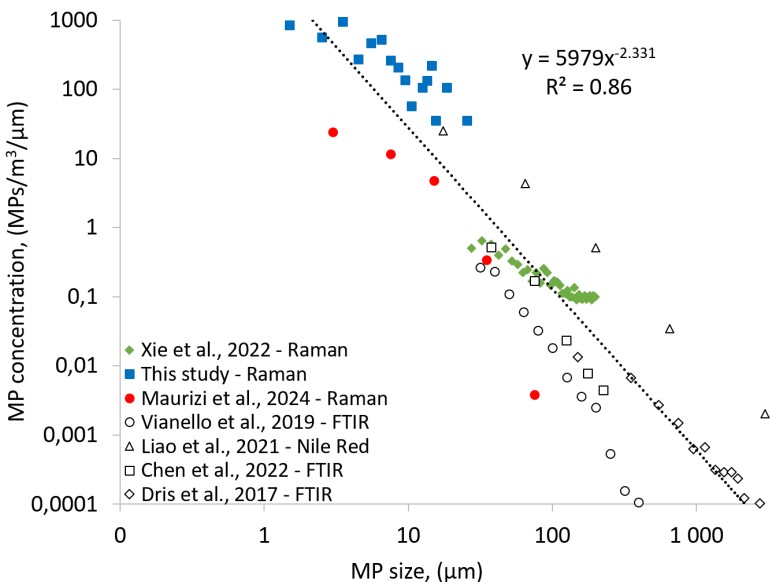

**Fig 4. Comparison of published indoor suspended MP concentrations in the 1–3,000 µm range.** FTIR microscopy typically probes MP > 20 µm, while Raman microscopy covers the MP$_{1-10\ \mu m}$ range down to 1 µm. The power law fit includes all data, except for select data points with low bias for small MP near the detection limits (inflected distributions from Vianello et al. [32] and Xie et al. [29]; see S9 Fig). The MP concentration variability reflects both true environmental and method variability and shows an overall coherent estimate of human airborne MP exposure, described by the equation y = 5979x$^{-2.331}$ (r² of 0.86), and for standardized 1µm wide bins, meaning that the function returns the MP concentration in the 1–2 µm range, for an x value of 1.5 µm.

concentration distribution in Fig 4. By integrating the area under the size distribution curve in Fig 4, we can calculate MP number concentrations for any given size interval in the 1–5,000 µm range:

$$C_{\tau_1-\tau_2} = \int_{\tau_1}^{\tau_2} A.x^b dx = \frac{A}{b+1} \cdot \left(\tau_2^{b+1} - \tau_1^{b+1}\right)$$

(3)

Where $C_{\tau_1-\tau_2}$ (MP m$^{-3}$) is the concentration (in number or mass respectively) of MP particles with size range $\tau_1$ to $\tau_2$ (µm). $A$ (MP m$^{-3}$ µm$^{-1}$) and $b$ are the intercept (5979) and the slope (−2.331) of the standardized particle size distribution respectively. $x$ is the MP size (µm). Table 1 summarizes the key human exposure metrics based on Fig 4, and suggests we inhale indoor air that contains on average 200 ± 180 MPs/m³ in the 10–300 µm range, and 4,300 ± 2,200 MPs/m³ in the 1–10 µm range. This indoor exposure estimate is based on mixed observations including occupational, residential and car cabin environments. It therefore represents an average human indoor MP exposure, and does not discern the factors that control local variability, such as interior surfaces (polymer or not), human activity (sleeping vs mobile), aeration etc. The new observation-based MP$_{1-10\ \mu m}$ concentration estimate of 4,300 MPs/m³ exceeds a previous extrapolated (from µFTIR data > 20 µm) estimate of 36 MPs/m³ by two orders of magnitude but lies within the 95th percentile (19,000 MPs/m³) of that study [40]. This illustrates the difficulty and large uncertainty associated with estimating small MP concentrations based on observations of larger MP by µFTIR. The new MP$_{1-10\ \mu m}$ estimate also exceeds our direct indoor MP$_{1-10\ \mu m}$ observation of 1,704 MPs/m³ (S4 Table), because the consensus power law in Fig 4 has a slightly higher MP abundance than our observations at the lower, 1–3 µm end, of the MP$_{1-10\ \mu m}$ range. Occupational exposure to airborne MP is pervasive in the textile industry, and multiple diseases have been reported among workers, including a threefold increase in lung cancer incidence among nylon flock workers [25]. The precise airborne MP exposure levels are not extensively documented, but

**Table 1. Estimates of indoor suspended MP fragment concentrations, based on all available literature observations in the 1.0 - 300 μm range shown in Fig 4 and fitted by the power law, $y = 5979x^{-2.331}$.** *Suspended NP concentrations in the 0.1–1.0 μm and 0.01–0.1 μm range are based on extrapolation of observed MP. Daily inhalation for children and adults is estimated based on concentrations and inhalation rates of $11 \pm 3$ and $16 \pm 4$ m³ per day respectively. All uncertainties are 1σ standard deviations. MP mass concentrations in picogram per m³ are estimated by assuming a MP density of 1 g/cm³, and an ellipsoidal MP fragment shape, and volume, $V = 0.1 \times D^3$, where D is MP diameter.*

| Size range | Concentration | | Inhalation, (particles/day) | |
|---|---|---|---|---|
| | (particles/m³) | (pg/m³) | Children | Adults |
| $MP_{10–300\ \mu m}$ | $200 \pm 180$ | 2.0E+08 | $2,200 \pm 2,000$ | $3,200 \pm 2,900$ |
| $MP_{1–10\ \mu m}$ | $4,300 \pm 2,200$ | 2.5E+02 | $47,000 \pm 28,000$ | $68,000 \pm 40,000$ |
| $NP_{0.1–1\ \mu m}$ | $94,000 \pm 48,000$ | 2.5E-01 | $1,000,000 \pm 600,000$ | $1,500,000 \pm 850,000$ |
| $NP_{0.01–0.1\ \mu m}$ | $2,100,000 \pm 1,200,000$ | 2.5E-10 | $23,000,000 \pm 15,000,000$ | $33,000,000 \pm 21,000,000$ |

have been cited to range from 10,000–1,000,000 fibers per m³ of air [41], which borders the indoor MP concentrations we observe.

Observations of atmospheric NP particles and MP in the same rainfall samples have been shown to extend the power law size distribution into the NP regime < 450 nm [42]. We therefore extrapolate the power law in Fig 4 to provide approximate estimates of potential human NP exposure by inhalation of $1,500,000 \pm 850,000$ NPs/m³ in the 0.1–1.0 μm range, and $33,000,000 \pm 21,000,000$ NPs/m³ in the 0.01–0.1 μm range. Indoor NP concentration observations are needed to verify these NP exposure estimates, especially because ultrafine particulate matter has a shorter atmospheric lifetime, leading to incorporation into larger aerosols (aggregation) or deposition to surfaces.

### Inhalation rates

For the evaluation of indoor human exposure through inhalation we use EU recommended default inhalation rates for two age groups including adults (31–51 years, $16 \pm 4$ m³/day) and children (2–12 years, $11 \pm 3$ m³/day) [35]. These default values provide a useful reference for harmonized exposure assessment and facilitate consistency across studies. However, actual inhalation rates can vary due to individual physiological differences, activity levels, exposure conditions, and other factors. We multiply these default inhalation rates by the consensus MP concentrations from Table 1, noting that the latter reflect a variety of indoor residential, occupational and car environments. Future studies should refine our understanding of the factors that control indoor MP concentrations, so that exposure estimates can be adapted to a wider spectrum of social and cultural lifestyles. The estimated MP inhalation rate of indoor $MP_{10–300\ \mu m}$ is $2,200 \pm 2,000$ and $3,200 \pm 2,900$ MPs/day for children and adults, respectively. These particles do not penetrate the lungs, but are most likely subjected to mucociliary clearance [43]. Inhaled MPs trapped in the mucus layer are moved from the lower respiratory tract (bronchi and bronchioles) to the upper respiratory tract (throat). When mucus with trapped MPs reaches the throat, it can be coughed up or cleared by the body through expectoration (spitting) or swallowing [44]. The process of swallowing is the most likely way to evacuate the mucus, but it leads to the transport of MPs to the gastrointestinal system. The potential $MP_{10–300\ \mu m}$ intestinal intake, from airborne exposure, exceeds best-estimates of direct dietary MP exposure of 553 and 858 MPs/day from food and beverages for children and adults in the 1–5,000 μm range [40], suggesting that inhalation of $MP_{10–300\ \mu m}$ is an indirect pathway for gastrointestinal MP exposure, adding to the overall burden of MPs in the human body.

The estimated inhalation of indoor suspended $MP_{1–10\ \mu m}$ is $47,000 \pm 28,000$ and $68,000 \pm 40,000$ MPs/day for children and adults, respectively (Table 1). Inhaled $MP_{1–10\ \mu m}$ can cross cellular barriers, entering the bloodstream and potentially causing systemic effects, including oxidative stress, immune responses, and even damage to vital organs over time [45]. Additionally, MPs can carry a range of toxic additives, including heavy metals and persistent organic pollutants, which may exacerbate their harmful effects [46,47]. These chemicals can leach out once inside the body, potentially disrupting endocrine functions [45], impairing cellular processes, or increasing the risk of cancer [24]. The combination of physical

and chemical stressors makes the inhalation of MPs particularly worrisome from a public health perspective. Finally, the potential inhalation of indoor suspended NP is 20 and 400-fold higher than for $MP_{1-10\ \mu m}$, yet requires observations for confirmation (Table 1).

In summary, our study documents that indoor suspended $MP_{1-10\ \mu m}$ concentrations are higher than previously thought. Consequently, human inhalation of fine particulate $MP_{1-10\ \mu m}$, and likely NP, that penetrate deep lung tissue may contribute to causing lung tissue damage, inflammation and associated diseases. We also suggest that inhaled $MP_{10-300\ \mu m}$ removed by mucociliary clearance, contributes to high intestinal MP intake. Future studies should aim for routine Raman analysis of $MP_{1-10\ \mu m}$ in order to investigate exposure control factors in detail, and include MP inhalation in epidemiological and occupational exposure studies.

## Supporting information

**S1 Table. Summary of indoor MP samples collected in residential and car cabin environments.**
(XLSX)

**S2 Table. Hit quality index (HQI) threshold applied to polymer identification by Raman microscopy.**
(XLSX)

**S3 Table. The spatial distribution of MP particles on the Anodisc filter.** Results were obtained from five spiked samples.
(XLSX)

**S4 Table. Summary of observed indoor suspended MP concentrations, given as median and interquartile range (IQR) for residential (n=7) and car cabin (n=5) environments.** Concentrations are reported for total and fine particulate matter (PM) size ranges from 1–10 μm ($PM_{10}$).
(XLSX)

**S1 Fig. Radial distribution of spiked red polyethylene (R-PE) particles from the center to the edge of the Anodisc filter.** Bins 1–4 correspond to concentric rings with mean distances from the center at 1.25, 3.75, 6.25, and 8.5 mm (S3 Table).
(TIF)

**S2 Fig. Raman spectrum of polydimethylsiloxane (PDMS) particle (blue) and reference spectrum of PDMS (red).**
(TIF)

**S3 Fig. Raman spectrum of polyethylene terephthalate (PET) particle (blue) and reference spectrum of PET (red).**
(TIF)

**S4 Fig. Raman spectrum of polypropylene (PP) particle (blue) and reference spectrum of PP (red).**
(TIF)

**S5 Fig. Raman spectrum of acrylonitrile butadiene styrene (ABS) particle (blue) and reference spectrum of ABS (red).**
(TIF)

**S6 Fig. Raman spectrum of polybutylene terephthalate (PBT) particle (blue) and reference spectrum of PBT (red).**
(TIF)

**S7 Fig. Raman spectrum of polyhydroxybutyrate (PHB) particle (blue) and reference spectrum of PHB (red).**
(TIF)

**S8 Fig.  Raman spectrum of polystyrene (PS) particle (blue) and reference spectrum of PS (red).**
(TIF)

**S9 Fig.  The relative abundance of indoor suspended MPs for different particle sizes.** Power law distributions are fitted for each study: $y = bx^{-\alpha}$ and have a mean exponent $\alpha$ of $-1.65 \pm 0.63(1\sigma)$.
(TIF)

## Acknowledgments

The authors would like to thank two anonymous reviewers for constructive comments and the editor for handling the manuscript.

## Author contributions

**Conceptualization:** Nadiia Yakovenko, Jeroen E. Sonke.

**Data curation:** Nadiia Yakovenko, Théo Segur, Jeroen E. Sonke.

**Formal analysis:** Nadiia Yakovenko, Lucía Pérez-Serrano, Théo Segur, Jeroen E. Sonke.

**Funding acquisition:** Jeroen E. Sonke.

**Investigation:** Nadiia Yakovenko, Lucía Pérez-Serrano.

**Methodology:** Nadiia Yakovenko, Lucía Pérez-Serrano, Oskar Hagelskjaer, Henar Margenat, Gaël Le Roux, Jeroen E. Sonke.

**Project administration:** Jeroen E. Sonke.

**Resources:** Jeroen E. Sonke.

**Software:** Jeroen E. Sonke.

**Supervision:** Jeroen E. Sonke.

**Validation:** Jeroen E. Sonke.

**Writing – original draft:** Nadiia Yakovenko, Lucía Pérez-Serrano, Jeroen E. Sonke.

**Writing – review & editing:** Nadiia Yakovenko, Théo Segur, Oskar Hagelskjaer, Henar Margenat, Gaël Le Roux, Jeroen E. Sonke.

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
