## [Decision Letter · Decision Letter 0]

Dear Dr. Yakovenko,

Thank you for submitting your manuscript to PLOS ONE. I have received the reports from our reviewers on your manuscript, PONE-D-25-08822 "Human exposure to PM10 microplastics in indoor air", which you submitted to PLOS ONE, and have decided that your manuscript can be reconsidered for publication should you be prepared to incorporate major revisions. Therefore, we invite you to submit a revised version of the manuscript that addresses the points raised during the review process. When preparing your revised manuscript, you are asked to carefully consider the reviewer comments which can be found below, and submit a list of responses to the comments.

We look forward to receiving your revised manuscript.

Kind regards,

Linton Munyai, PhD

Academic Editor

PLOS ONE

 [ANR-20-CE34-0014 ATMO-PLASTIC

ANR-23-CE34-0012 BUBBLPLAST].

At this time, please address the following queries

**Additional Editor Comments:**

Dear Authors,

I would like you to clarify few things on the manuscript:

1. The height of 1.6-1.7m, can you please justify the selection of this height?

2. In cars, how did you standardize the collection of air particles, Was the air-condition always on/off? During the period of collection, were the windows closed or opened? I assume if windows are open and aircorn is on, there will be re-suspension of particles inside the car. How did you standardize that from one car to the other? Same applies inside the house. When sweeping, whether the floor is carpet or tiled, there will also be re-suspension. How was all these accounted for?

3. I missed a point on verification of whether it was plastics inhaled, or it is other dust particles? Can you put readers into confidence on how microplastics were verified.

Reviewers' comments:

**Comments to the Author**

**Reviewer #1: **

This manuscript is an interesting and important contribution to the study of airborne microplastics. The topic is highly relevant, and the findings add valuable insights into indoor exposure, which makes it even more compelling. The study is well thought out, and the methods are strong.

The comments below are just suggestions to help improve clarity and readability. The research is already solid, and with a few changes, it will be even stronger and easier for readers to engage with.

1) Lines 24–26: The abstract presents interesting findings, but the take-home message could be clearer. Consider emphasizing why your study is particularly significant compared to previous work. For example, rephrase the last two sentences to highlight the critical health implications more directly.

2) Line 20: The term "power size distribution law" is mentioned in the abstract but not explained. Since this may not be familiar to all readers, a brief clarification (even a parenthetical note) would improve accessibility.

3) Lines 72–78: The inclusion of the car cabin environment is an interesting and novel aspect, but its significance isn’t immediately clear. Would it be useful to briefly explain why car environments were chosen alongside residential space? perhaps highlighting their relevance to daily exposure?

4) Lines 62–65: The introduction covers a lot of ground effectively, but the transition to the research gap could be smoother. The sentence starting with "However, despite its potential, only two studies [27,28]..." feels abrupt. Could you introduce this section with a more gradual lead-in?

5) Lines 14, 20, 68 .. : The manuscript switches between "MP1–10µm" and "MP1–10 µm" (with and without a space). Standardizing this across the text will improve readability.

6) Lines 88–90: The choice of sampling heights (1.6–1.7 m for living rooms and 0.5 m for bedrooms) makes sense intuitively but would benefit from a brief justification. Does this reflect average breathing heights?

7) Lines 188–192: the author mentions that blank corrections were applied, but it would be helpful to explain if any contamination patterns were observed across different polymer types. Did specific polymers appear more frequently in blanks, and if so, how was this addressed?

8) Lines 320–322: the author reports that MP concentrations were higher in cars than in apartments but also mention that this difference was not statistically significant (p = 0.5). Perhaps rewording to avoid implying a strong contrast would prevent misinterpretation.

9) Lines 366–369: The estimated inhalation rates are striking, but a brief discussion of uncertainty sources (e.g., variability in human breathing rates, exposure conditions) would help contextualize the robustness of these values.

10) Lines 272–276, 300–301: The figures (1,2,3 and 4) are informative, but are the labels and axis text large enough for clear readability? I suggest improving the image quality by using higher-resolution versions.

11) Lines 317–319: The discussion highlights how your estimates are higher than previous extrapolations, which is great. However, a sentence explaining why earlier estimates were lower (e.g., due to detection limits, methodology differences) would provide helpful context.

12) Lines 348–350: The study focuses on indoor exposure, but some readers may wonder how these findings compare to occupational environments with high plastic use (e.g., textile industries). A brief mention of this in the discussion could add depth.

13) Lines 381–384: The potential health impacts of inhaling MP1–10µm are mentioned, but could you add one or two specific examples of known biological effects (e.g., inflammation, oxidative stress) to make this more concrete?

14) Lines 395–398: The conclusion does a great job summarizing findings, but a forward-looking statement about next steps in this field (e.g., improving detection methods, longitudinal exposure studies) would make for a stronger closing.

**Reviewer #2:**  

• The study addresses a critical and emerging issue—human exposure to airborne microplastics, particularly in the PM10 range, which is often overlooked.

• The use of Raman spectroscopy enhances the reliability of microplastic identification and size classification down to 1 µm.

• The study successfully compares indoor MP concentrations across different environments (residential vs. car cabin), contributing to the understanding of MP inhalation risks.

• The Introduction effectively highlights the importance of studying airborne microplastics, but a clearer transition into study objectives would improve flow.

• The Discussion could be streamlined to avoid redundancy (e.g., repeated explanations about particle inhalation pathways).

• The Figures & Tables are valuable, but better integration into the text with clear references to their significance would be helpful.

• It would be beneficial to provide more details about sampling duration and how variability in MP concentrations was controlled.

• Clarify whether cross-contamination controls were performed beyond blanks (e.g., during sampling in the field).

• Some comparisons, such as MP concentrations between apartments and cars, use median values. A statistical significance test (e.g., Mann-Whitney U test) should be explicitly mentioned.

• The power law fitting for MP size distribution is informative, but additional explanation about its implications for human exposure would be useful.

• The study makes strong claims about increased MP inhalation leading to health risks. While plausible, direct toxicological evidence linking MP1-10µm to specific diseases should be elaborated with relevant citations.

• Consider discussing how long-term exposure may differ from acute exposure in different environments.

• Minor grammatical errors and awkward phrasing should be polished. Example: "Given these findings, and the fact that people spend 90% of their time indoors, the greater potential for exposure to MPs through inhalation in indoor environments should be emphasized." Suggested revision: "Given that people spend 90% of their time indoors, the potential for inhalation exposure to MPs in indoor environments is significantly higher and warrants attention."

• The study provides valuable insights into indoor airborne microplastics; however, several recent studies on ambient microplastics have been published. I recommend the authors conduct a thorough keyword search to identify and incorporate relevant literature. Strengthening the discussion by comparing indoor and ambient microplastic studies will enhance the manuscript's impact and provide a more comprehensive context for human exposure risks.

• Line 35 marine boundary layer it would be good if you will rewrite this sentence as Over the past decade, MPs have been detected in outdoor atmospheric aerosols [5–8] and deposition [9–14], in various parts of the world, from urban and highly industrialized areas [9,13] to remote mountainous regions [10,12], the marine boundary layer [7] and indoor environments. As it will give proper flow For indoor environment cite following references. Unravelling the microplastic contamination: a comprehensive analysis of microplastics in indoor house dust. An in-depth study of dust samples reveals microplastic (MP) contamination in indoor commercial markets.

• Lines 62 and 63 check the below articles as Raman has been used in indoor PM microplastics studies. So cite them here as it is contradictory when there is a gap. Airborne microplastic contamination across diverse university indoor environments: A comprehensive ambient analysis. A comprehensive characterization of indoor ambient microplastics in households during the COVID-19 pandemic. A comprehensive characterization of indoor ambient microplastics in households during the COVID-19 pandemic.

• Line 303 check the missing literature. You can cite the above literature here also.

• Compare your results of lnhaltion rates with the recent studies like. Airborne microplastic contamination across diverse university indoor environments: A comprehensive ambient analysis.. A comprehensive characterization of indoor ambient microplastics in households during the COVID-19 pandemic…

---

## [Author Response · Author response to Decision Letter 1]

6 Jun 2025

We thank the reviewers and editor for their constructive comments and questions. Below, point by point replies are provided to each comment. All changes have been made accordingly in the revised manuscript.

Response: We have added clarification to the Materials and Methods section on Line 85 (L85): “No permits were required because the residential and car environments studied are all privately owned by the authors of this study.”

[ANR-20-CE34-0014 ATMO-PLASTIC

ANR-23-CE34-0012 BUBBLPLAST].

At this time, please address the following queries

Response: We added a Funding section on L455 where we acknowledge our funders: “This study, including the salaries of NY and HM, was supported by ANR-20-CE34–0014 “ATMO-PLASTIC” and ANR-23-CE34-0012 “BUBBLEPLAST” grants from the French Agence Nationale de Recherche. OH was funded by the 80Prime CNRS «4DμPlast» PhD fellowship, and TS by a PhD fellowship from the French ministry of education.”

Additional Editor Comments:

Dear Authors,

I would like you to clarify few things on the manuscript:

1. The height of 1.6-1.7m, can you please justify the selection of this height?

Response: For sampling in the residential environment, the filter holders were placed horizontally at a height of 1.6–1.7 m from the floor, corresponding to the average human inhalation height (Liao et al., 2021). We added clarification to the text on L102–106: “For sampling in the residential environment, the filter holders were placed horizontally at a height of 1.6–1.7 m above the floor in living rooms, corresponding to the average human inhalation height [18]. In bedrooms, a height of 0.5 m was chosen to represent the average inhalation height during sleep, based on the bed height in the studied bedrooms.”

2. In cars, how did you standardize the collection of air particles, Was the air-condition always on/off? During the period of collection, were the windows closed or opened? I assume if windows are open and aircorn is on, there will be re-suspension of particles inside the car. How did you standardize that from one car to the other? Same applies inside the house. When sweeping, whether the floor is carpet or tiled, there will also be re-suspension. How was all these accounted for?

Response: Sampling in both cars was carried out while driving the car with the windows closed, AC off, and outside air entering the cabin through vents at a regular rate. In the residences, windows were closed. Some of this was mentioned, but we added further clarification to the Methods section.

3. I missed a point on verification of whether it was plastics inhaled, or it is other dust particles? Can you put readers into confidence on how microplastics were verified.

Response: We quantified the concentrations of suspended (floating) MPs in indoor air. The MP identification (verification) was made by Raman microscopy, and example Raman spectra for MP are illustrated in the Supporting Information. The observed MP make up a fraction of total dust particles. We did not quantify the other dust particle concentrations. The estimate of inhaled MP is a calculation that is based on the observed suspended MP and the typical human daily air intake into the lungs.

We have made changes at the end of the introduction to improve flow and clarity on our motivation and objectives.

Reviewers' comments:

Reviewer #1:

This manuscript is an interesting and important contribution to the study of airborne microplastics. The topic is highly relevant, and the findings add valuable insights into indoor exposure, which makes it even more compelling. The study is well thought out, and the methods are strong.

The comments below are just suggestions to help improve clarity and readability. The research is already solid, and with a few changes, it will be even stronger and easier for readers to engage with.

Response: Thank you for the positive feedback.

1) Lines 24–26: The abstract presents interesting findings, but the take-home message could be clearer. Consider emphasizing why your study is particularly significant compared to previous work. For example, rephrase the last two sentences to highlight the critical health implications more directly.

Response: We rephrased the concluding phrase on L24 as follows: “The MP1–10 µm exposure estimates are 100-fold higher than previous estimates that were extrapolated from larger MP sizes, and suggest that the health impacts of MP inhalation may be more substantial than we realize.”

2) Line 20: The term "power size distribution law" is mentioned in the abstract but not explained. Since this may not be familiar to all readers, a brief clarification (even a parenthetical note) would improve accessibility.

Response:We added the clarification: “…following a power size distribution law (the number of MP fragments increases exponentially as particle size decreases).”

3) Lines 72–78: The inclusion of the car cabin environment is an interesting and novel aspect, but its significance isn’t immediately clear. Would it be useful to briefly explain why car environments were chosen alongside residential space? perhaps highlighting their relevance to daily exposure?

Response: We added a comment early on in the introduction section, on L44, stating that “people spend approximately 5% (Mannan et al., 2021) of their time in a vehicle.”

We also added further motivation to the Materials and Methods section on L89: “Car cabin indoor environment differs from residential indoor environments, such as homes, offices, and commercial buildings and has a different microclimate. Car cabins are smaller, enclosed and mostly made of synthetic materials such as plastic, which can be potential sources of MPs.”

4) Lines 62–65: The introduction covers a lot of ground effectively, but the transition to the research gap could be smoother. The sentence starting with "However, despite its potential, only two studies [27,28]..." feels abrupt. Could you introduce this section with a more gradual lead-in?

Response: We tried to follow your advice and reorganized this section.

5) Lines 14, 20, 68 .. : The manuscript switches between "MP1–10µm" and "MP1–10 µm" (with and without a space). Standardizing this across the text will improve readability.

Response: Sorry about this. We have checked and corrected the issue throughout the manuscript by choosing the format with space (e.g., MP1–10 µm).

6) Lines 88–90: The choice of sampling heights (1.6–1.7 m for living rooms and 0.5 m for bedrooms) makes sense intuitively but would benefit from a brief justification. Does this reflect average breathing heights?

Response: The editor made the same comment; we added clarifications on L102-106: “For sampling in the residential environment, the filter holders were placed horizontally at a height of 1.6–1.7 m above the floor in living rooms, corresponding to the average human inhalation height (Liao et al., 2021). In bedrooms, a height of 0.5 m was chosen to represent the average inhalation height during sleep, based on the bed height in the studied bedrooms.”

7) Lines 188–192: the author mentions that blank corrections were applied, but it would be helpful to explain if any contamination patterns were observed across different polymer types. Did specific polymers appear more frequently in blanks, and if so, how was this addressed?

Response: We added information on the analysis of blank content to the Results section on L265–269: “A total of five types of polymers were detected in the four blanks, including PE, PP, PET, PHB, and PDMS. Although some polymers, such as PE, PET, and PP, were observed in multiple blanks, the detected MPs varied in both polymer types and size range. Overall, no consistent pattern of contamination was observed among the blanks.”

As mentioned on L205–209: “Polymers found in blanks were subtracted from their corresponding samples based on chemical composition and diameter range (e.g., if the blank presented two particles reported as PE in a [1,2) μm diameter range, these particles were subtracted from the count of PE [1,2) μm class in the sample).

8) Lines 320–322: the author reports that MP concentrations were higher in cars than in apartments but also mention that this difference was not statistically significant (p = 0.5). Perhaps rewording to avoid implying a strong contrast would prevent misinterpretation.

Response: We reformulated our phrase to be more clear, L343: “Although the median MP concentration in the car cabin environment was higher than in apartments, this difference was not statistically significant (Mann-Whitney U test; p-value = 0.5) due to the high variability in both environments.”

9) Lines 366–369: The estimated inhalation rates are striking, but a brief discussion of uncertainty sources (e.g., variability in human breathing rates, exposure conditions) would help contextualize the robustness of these values.

Response: Breathing rate variability is included in the inhalation uncertainty; exposure conditions (car, office, human movement, surface polymers present, cleaning frequency, ventilation, population) are likely the larger source of variability. We find it premature at this stage to attempt to incorporate exposure variability in the estimate; this is the reason we combine all the exposure variability observed in the literature and in this study into a single broad human exposure estimate. We added context to L398-405: “These default values provide a useful reference for harmonized exposure assessment and facilitate consistency across studies. However, actual inhalation rates can vary due to individual physiological differences, activity levels, exposure conditions, and other factors. We multiply these default inhalation rates by the consensus MP concentrations from Table 1, noting that the latter reflect a variety of indoor residential, occupational and car environments. Future studies should refine our understanding of the factors that control indoor MP concentrations, so that exposure estimates can be adapted to a wider spectrum of social and cultural lifestyles.”

10) Lines 272–276, 300–301: The figures (1,2,3 and 4) are informative, but are the labels and axis text large enough for clear readability? I suggest improving the image quality by using higher-resolution versions.

Response: It appears that the automatic creation of PDFs on the PLOS One platform after submission reduces the quality of images, making them appear blurry. However, all images were created using high-resolution versions, following the journal’s requirements: TIFF format, less than 10MB in size, and 300–600 dpi. We will try to improve the text size and possibly increase the resolution > 300 dpi…

11) Lines 317–319: The discussion highlights how your estimates are higher than previous extrapolations, which is great. However, a sentence explaining why earlier estimates were lower (e.g., due to detection limits, methodology differences) would provide helpful context.

Response: This is slightly confusing. The Line numbers 317-319 are not associated with our text on comparing our new MP concentration estimate with the previous ‘extrapolated’ estimate; the correct old line numbers are 349-352. We added a phrase on new L377 (in blue here): “The new observation-based MP1–10µm concentration estimate of 4,300 MPs/m3 exceeds a previous extrapolated (from µFTIR data > 20 µm) estimate of 36 MPs/m3 by two orders of magnitude but lies within the 95th percentile (19,000 MPs/m3) of that study [37]. This illustrates the difficulty and large uncertainty associated with estimating small MP concentrations based on observations of larger MP by µFTIR.”

12) Lines 348–350: The study focuses on indoor exposure, but some readers may wonder how these findings compare to occupational environments with high plastic use (e.g., textile industries). A brief mention of this in the discussion could add depth.

Response: We added the following discussion on L381, citing two influential reviews: “Occupational exposure to airborne MP is pervasive in the textile industry, and multiple diseases have been reported among workers including a threefold increase in lung cancer incidence among nylon flock workers (Landrigan et al., 2023). The precise airborne MP exposure levels are not extensively documented, but have been cited to range from 10,000 to 1,000,000 fibers per m3 of air (Wright & Kelly, 2017), which borders the indoor MP concentrations we observe.”

13) Lines 381–384: The potential health impacts of inhaling MP1–10µm are mentioned, but could you add one or two specific examples of known biological effects (e.g., inflammation, oxidative stress) to make this more concrete?

Response: We expanded the discussion on L417: “Inhaled MP1–10µm can cross cellular barriers, entering the bloodstream and potentially causing systemic effects, including oxidative stress, immune responses, and even damage to vital organs over time (Revel et al., 2018). Additionally, MPs can carry a range of toxic additives, including heavy metals and persistent organic pollutants, which may exacerbate their harmful effects (Thompson et al., 2009). These chemicals can leach out once inside the body, potentially disrupting endocrine functions (Revel et al., 2018), impairing cellular processes, or increasing the risk of cancer (Prata et al., 2018). The combination of physical and chemical stressors makes the inhalation of microplastics particularly worrisome from a public health perspective.”

14) Lines 395–398: The conclusion does a great job summarizing findings, but a forward-looking statement about next steps in this field (e.g., improving detection methods, longitudinal exposure studies) would make for a stronger closing.

Response: Good idea; we added a closing phrase on L442: “Future studies should aim for routine Raman analysis of MP1–10 µm in order to investigate exposure control factors in detail, and include MP inhalation in epidemiological and occupational exposure studies.”

Reviewer #2:

1. The study addresses a critical and emerging issue—human exposure to airborne microplastics, particularly in the PM10 range, which is often overlooked.

2. The use of Raman spectroscopy enhances the reliability of microplastic identification and size classification down to 1 µm.

3. The study successfully compares indoor MP concentrations across different environments (residential vs. car cabin), contributing to the understanding of MP inhalation risks.

4. The Introduction effectively highlights the importance of studying airborne microplastics, but a clearer transition into study objectives would improve flow.

Response: Thank you for the encouraging feedback; we have made multiple changes to the introdu

---

## [Decision Letter · Decision Letter 1]

Human exposure to PM10 microplastics in indoor air

PONE-D-25-08822R1

Dear Dr. Yakovenko,

We’re pleased to inform you that your manuscript has been judged scientifically suitable for publication and will be formally accepted for publication once it meets all outstanding technical requirements.

Kind regards,

Linton Munyai, PhD

Academic Editor

PLOS ONE

Additional Editor Comments (optional):

Reviewers' comments:

Reviewer's Responses to Questions

**Comments to the Author**

Reviewer #1: All comments have been addressed

Reviewer #2: All comments have been addressed

2. Is the manuscript technically sound, and do the data support the conclusions?

Reviewer #1: Yes

Reviewer #2: Yes

3. Has the statistical analysis been performed appropriately and rigorously?

Reviewer #1: Yes

Reviewer #2: Yes

4. Have the authors made all data underlying the findings in their manuscript fully available?

Reviewer #1: Yes

Reviewer #2: Yes

5. Is the manuscript presented in an intelligible fashion and written in standard English?

Reviewer #1: Yes

Reviewer #2: Yes

Reviewer #1: I think this revised manuscript ’Human exposure to PM10 microplastics in indoor air’ is interesting and relevant, and I appreciate the effort the authors have put into addressing the previous comments. Their responses were clear and constructive, and the corresponding changes have improved the overall quality and clarity of the manuscript.

I am satisfied with the revisions and have no further comments. I believe the manuscript is now suitable for publication in its current form.

Reviewer #2: (No Response)

**Do you want your identity to be public for this peer review?** For information about this choice, including consent withdrawal, please see our Privacy Policy

Reviewer #1: No

Reviewer #2: No

---

## [Editor Report · Acceptance letter]

PONE-D-25-08822R1

PLOS ONE

Dear Dr. Yakovenko,

I'm pleased to inform you that your manuscript has been deemed suitable for publication in PLOS ONE. Congratulations! Your manuscript is now being handed over to our production team.

Kind regards,

on behalf of

Dr. Linton Munyai

Academic Editor

PLOS ONE